# Experimental Determination of Pedestrian Thermal Comfort on Water-Retaining Pavement for UHI Adaptation Strategy

**Yasuhiro Shimazaki** [1,2,*], **Masashige Aoki** [3], **Jumpei Nitta** [2,3], **Hodaka Okajima** [3] and **Atsumasa Yoshida** [4]

1   Department of Architecture and Civil Engineering, Toyohashi University of Technology,
    Toyohashi 441-8580, Japan
2   Department of Human Information Engineering, Okayama Prefectural University, Soja 719-1197, Japan;
    jumpei_nitta@taiseirotec.co.jp
3   Taisei Rotec Corporation, Konosu 365-0027, Japan; masashige_aoki@taiseirotec.co.jp (M.A.);
    hodaka_okajima@taiseirotec.co.jp (H.O.)
4   Department of Mechanical Engineering, Osaka Prefecture University, Sakai 599-8531, Japan;
    ayoshida@me.osakafu-u.ac.jp
*   Correspondence: shimazaki@ace.tut.ac.jp; Tel.: +81-532-44-6838

**Abstract:** Artificial impervious surfaces are one of the most significant factors contributing to urban heat islands (UHIs). Adapting to UHIs is a challenge in achieving thermal comfort. We conducted a quantitative and subjective evaluation of a closely paved novel water-retaining pavement (WR) and a conventional dense-asphalt pavement (AS). We investigated the thermal states of humans based on the human energy balance known as "human thermal load" as an indicator for the assessment, and the original human thermal load method was improved for assessing nonuniform environments such as pavements. We looked for individual thermal perceptions simultaneously. The experiment was conducted in typical summer weather. The surface temperature of the WR was found to be significantly lower, by 9.5 °C, while the air temperature and humidity above both pavements were not significantly different. Thus, air conditions did not directly affect the sensible and latent heat loss. The reflected solar radiation was significantly larger, and the infrared radiation was significantly smaller on the WR than on the AS due to the lower surface temperature from the water evaporation and higher reflectance. Further, the human thermal load at a pedestrian level of 1.5 m was found to be significantly different: 237 W/m$^2$ for AS and 215 W/m$^2$ for WR. In a subjective evaluation, the perceptions of WR tend to be distributed in smaller human thermal load, thereby resulting in a cooler and comfortable sensation. Therefore, we demonstrated that when compared to AS, WR significantly improves thermal comfort.

**Keywords:** human thermal load; surface material; evaporation; watering; subjective experiment

## 1. Introduction

Climate change is a fundamental problem of our time. Like other countries or regions, Japanese cities have been setting new records for the highest temperatures almost every year because of the climate warming trend and urban heat islands (UHIs). UHIs lead to high energy consumption and the deterioration of the quality of life in densely populated areas. This scenario is linked to the outdoor thermal comfort condition because the level of comfort is an essential factor for promoting outdoor activity, especially in urban areas.

It is well known that different types of ground-surface covers affect the climate of the built environment. The thermo-physical properties of surfaces in cities (e.g., asphalt) result in modified surface radiation and heat balance. In addition, pavements account for 20–40% of the surface area of a typical city, and therefore, they have the potential to mitigate UHIs [1]. Air temperatures depend on the absorption of solar radiation. Conventional absorptive pavements become hotter because of their higher absorption or lower albedo, and the hotter pavement warms the surrounding air. In contrast, natural ground contains

water, which suppresses the rise in its temperature. Artificial impervious surfaces are considered one of the most significant causes of UHIs. The hot pavements aggravate UHIs by warming the surrounding air.

Temperature-lowering functional "cool pavements," which use novel materials and design modifications, can provide cooling compared to conventional pavements. Cool pavements can be categorized into two categories: reflective and evaporative [2]. Cool-pavement technologies have been developed mostly by enhancing the surface reflectivity, owing to the cost and ease of installation [3]. Because the solar heat absorbed by the pavement results in a rise in surface temperature, the pavement temperature could be decreased by decreasing surface absorptance (by increasing the reflectance). An approximately 20 °C difference in surface temperature was reported between black-colored and white-colored pavements during summer experiments [4]. Increasing reflectance by 0.1 could be analytically estimated to reduce the maximum pavement surface temperature by about 3.3 °C [5]. Researchers in Japan have been diligently working on water-retaining pavements. Because the energy is taken during the evaporation process, evaporative pavements contribute to cooling of the pavement surfaces. Temperature reductions of up to about 20 °C compared to conventional pavements have been reported [6–9]. The Japanese have a traditional "Uchimizu" watering custom, in which people sprinkle leftover water, such as used bath water, on the streets on hot summer days to create a cooling effect from evaporation. Similarly, water-retaining-pavement technology is believed to be environmentally friendly because it uses frequent natural rainfall effectively. In order to evaluate the effectiveness and the applicability of evaporative pavements, verification experiments were conducted in real outdoor environments.

Thermal-comfort studies in Japan are extensive, because achieving thermal comfort is a challenge due to the hot and humid summers in the country. These studies are mostly based on the measurement of physical environmental quantities. Because the sun represents the main heat source outdoors and influences the outdoor thermal environment directly or indirectly, assessing radiative components is important. Heat mitigation in pavement studies tends to focus on changes in surface temperature. This is only one of the factors that influences human thermal comfort. Few studies have focused on the effects of human aspects on functional evaporative pavements and how humans respond to these pavements. Although outdoor thermal indices such as the physiological equivalent temperature (PET) [10] and the universal thermal climate index (UTCI) [11] have become increasingly common, existing thermal indices developed for indoor conditions, such as the standard effective temperature (SET*) and the predicted mean vote (PMV), have been applied to outdoor conditions directly or with modifications [12]. Thermal comfort can be predicted using two different methods: the rational model and the adaptive model [13–15]. The rational model is based on human energy balance, and the adaptive model is based on thermal adaptation. To predict the thermal comfort conditions accurately, it is desirable to consider both aspects simultaneously. It is necessary to understand the subjective performance through an experimental approach.

Pedestrian spaces in cities are public spaces, and they play an important role in the physical and mental well-being of pedestrians. Although a variety of UHI mitigation and adaptation evaluation measures are already present, understanding how humans interact with their environment is fundamentally important for developing an evaluation tool. An experimental assessment method for pedestrian comfort is proposed here to provide a comfortable urban space using a functional evaporative pavement. The thermal environment on a water-retaining pavement and on a conventional asphalt pavement was investigated with Japanese participants.

## 2. Experiments

### 2.1. Material

Test pavements were constructed at Okayama Prefectural University in Okayama. The campus is located at 34°41′32″ N, 133°46′54.5″ E in Japan's western region, along the

Seto Inland Seacoast. Okayama is known as the "land of sunshine" because of its minimal rain and mild climate throughout the year. Previously, evaporative cooling duration for the WR was reported to be a few days to one week [2]. Based on the data from the Japan Meteorological Agency, the average rainfall in Okayama during summer is 171 mm and 17 days for June, 160 mm and 17 days for July, 87 mm and 13 days for August, and 134 mm and 17 days for September; for practical applications, the amount of rainfall is considered sufficient for watering. The location of the test was an open space. The pavement materials used in the experiment were WR and AS. The WR pavement was made of water-retentive blocks composed of glass (10%) and ceramic (90%), and thus, it contained fine voids to hold water (water retention capacity $\geq$ 15 vol%). The general dimensions of the water-retentive block were: length $\times$ width = 100 mm $\times$ 200 mm, and thickness = 60 mm. Both pavements had an area of 7.0 m $\times$ 7.0 m, as shown in Figure 1.

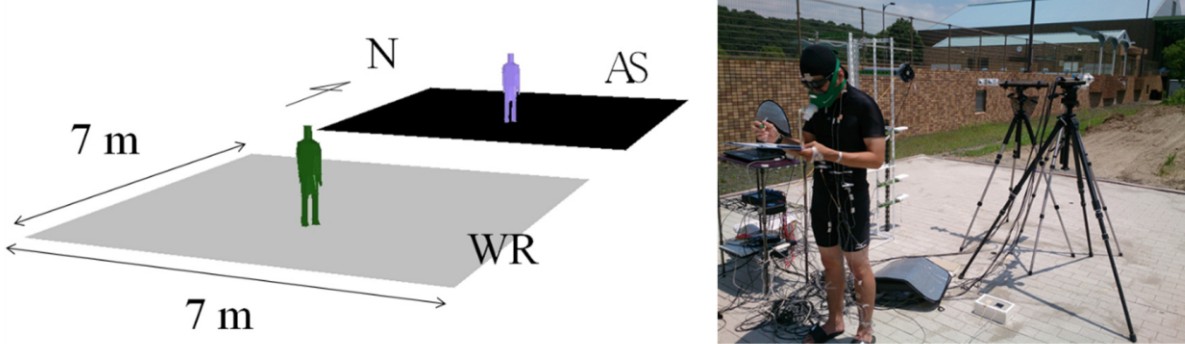

**Figure 1.** The experimental landscape.

*2.2. Methods*

2.2.1. Experimental Setup

We measured the physiological and psychological reactions of the human body to different environmental conditions due to the pavements. Further, the surrounding weather factors were also measured. The experiment was conducted between 10:00 a.m.–16:00 p.m. JST from 22 August to 26 August 2016.

A total of 14 healthy male Japanese university students participated in the study for both pavements. Only one participant stayed on each pavement at a time, and the experiment was conducted on both pavements simultaneously. The height, weight, and age of volunteers were 172.7 $\pm$ 4.5 cm, 63.4 $\pm$ 6.6 kg, and 22.4 $\pm$ 1.3 years, respectively. The body mass index (BMI) of the participants was 21.1 $\pm$ 1.7, which was categorized in the healthy weight range. Informed consent was obtained from all volunteers, and the research was conducted in accordance with the guidelines and approval of the research ethics committee of the institute.

Each trial lasted for 30 min, during which the participants maintained a standing-still posture. To ensure consistent clothing insulation, each participant dressed in the same tight-fit garments and footwear; the properties of these garments were determined before the trial (whole clothing insulation I = 0.35 clo). Each participant drank 200 mL of water 30 min before the experiment to ensure suitable hydration.

2.2.2. Climatic Observation

Physical environmental quantities including air temperature, humidity, solar and infrared radiation, and wind speed were measured. Air temperature and humidity were measured at heights of 0.35 m, 0.5 m, and 1.5 m from the surface using a capacitive thermo-hygrometer recorder (TR-73U, T&D Corporation, Nagano, Japan). The solar radiation and infrared radiation in upward and downward directions were measured at a height of 1.5 m using a net radiometer (MR-60, EKO, Instruments, Tokyo, Japan). The wind speed was

measured at a height of 1.5 m using a hot-wire anemometer (Climomaster Model-6531, Kanomax Japan Inc., Tokyo, Japan). A measurement height of 1.5 m was selected to indicate the pedestrian height [12,16]. To understand the effect of the height on the environment, near-surface values at 0.35 and 0.5 m were measured for temperature and humidity. Every environmental quantity was measured and recorded at 1 min intervals using a data-logger (LR-8400, Hioki E.E. Corp., Nagano, Japan).

### 2.2.3. Pavement Conditions

The surface temperatures at five points on both pavements were measured using J-type thermocouples. A thermocouple was placed at the center of the pavement, and four thermocouples were placed 0.5 m away from the center of the pavement in the north, east, south, and west directions. Infrared images were periodically captured during the experiment using thermography (InfReC H2640, Nippon Avionics Co., Yokohama, Japan). The emissivity was set to 1.0 to obtain infrared images for both pavements.

The degree of reflectance (albedo) is an important factor in the formation of urban climate [17]. Because the experimental pavements were relatively large and ensured uniformity, the values of albedo $\rho$ were obtained simply as the ratio of the amount of global solar radiation $S\downarrow$ and the reflected solar radiation $S\uparrow$ measured by the net radiometer [18].

$$\rho = \frac{S\uparrow}{S\downarrow} \tag{1}$$

A water hose was used to sprinkle water until the WR blocks could not retain water any more to ensure uniformity in the conditions before each experiment.

### 2.2.4. Human Thermal States on Pavements

To design a better environment for humans by improving ground coverings, we investigated and quantified the thermal states of humans on pavements. We previously developed a measure for evaluating human thermal environment outdoors known as "human thermal load" [19]; this measure is based on human energy balance. When the human body and the surrounding environment are in a state of thermal equilibrium, the thermal condition of the human body can be expressed by the heat balance as:

$$M + R_{net} = W + C + E \tag{2}$$

where $M$, $R_{net}$, $W$, $C$, and $E$ denote the metabolic rate, net radiation, workload, convective heat loss, and evaporative heat loss, respectively. Further, $C$ and $E$ include the heat exchange caused by respiration. The unit for each term is W/m$^2$.

If the thermal state is not at a neutral level, a positive or negative thermal load is applied to the human body. This load amount is referred to as the human thermal load $F_{load}$ (W/m$^2$) and is defined by the heat balance equation as:

$$F_{load} = M + R_{net} - W - C - E \tag{3}$$

Metabolic rate is an important determinant of the comfort or the strain resulting from the exposure to thermal environment, particularly in a hot climate [20]. The metabolic rate $M$ refers to the heat generation by humans, and it is calculated from body surface area $A_D$ (m$^2$), oxygen consumption $V_{O_2}$ (L/min), and carbon-dioxide elimination $V_{CO_2}$ (L/min) using the metabolic measurement system (VO2000, MGC Diagnostics, Saint Paul, MN, USA) based on Weir's formula [21].

$$M = \frac{69.735\left(3.9V_{O_2} + 1.1V_{CO_2}\right)}{A_D} \tag{4}$$

The body surface area $A_D$ can be determined using the following formula by Kurazumi et al. [22] for Japanese participants:

$$A_D = 2034.309W^{0.425}H^{0.725} \tag{5}$$

where $W$ (kg) denotes the body weight and $H$ (m) denotes the height.

The metabolic rate can be determined according to the type of activity and occupant. As a simplified method for outdoor experiments for practical applications, the metabolic equivalents of task (met) can be used for determining the activity level. The met is defined as the ratio of the working metabolic rate relative to the resting metabolic rate for an activity, and the list of met values for different activities is widely available (e.g., ASHRAE) [23]. To reflect individual variability, the resting metabolic rate is preliminary determined in an indoor chamber using Equation (4), and then, the metabolic rate is determined by multiplying with the met value. The met value for a person standing still is met = 1.2.

Workload $W$ denotes the mechanical work performed by humans. Since the human participants were standing and in a resting posture in this experiment, it was assumed that $W = 0$ [23].

The net radiation $R_{net}$ is the amount of solar and infrared radiation received by the human body and emitted from the human body, and it is calculated as:

$$R_{net} = (1 - \alpha_h)R_{sh} + \varepsilon_h\left\{R_{ln} - \sigma(T_{skin} + 273.15)^4\right\} \tag{6}$$

where $\alpha_h$, $\varepsilon_h$, $R_{sh}$ (W/m$^2$), $R_{ln}$ (W/m$^2$), $\sigma$ (W/(m$^2$K$^4$)), and $T_{skin}$ (°C) denote the reflectance of the human body (=0.3) [24], emissivity of the human body (=0.98) [25], gain of heat from solar radiation, gain of heat from infrared radiation, the Stefan–Boltzmann constant, and the mean skin temperature, respectively. As the participants were dressed in a tight-fit garment in this experiment, $\alpha_h$ can be partly replaced by the reflectance of the garment $\alpha_{clo}$, which was preliminarily determined using the method proposed by the authors [18]. The mean skin temperature was the area weighted and calculated as:

$$T_{skin} = \sum_i F_i T_i \tag{7}$$

where $i$, $F_i$, and $T_i$ (°C) denote the body region, weighting factor for region $i$, and skin temperature for region $i$, respectively. The measuring sites varied from using fewer point to a large number of points, such as Ramanathan 4-points, Hardy-DuBois 7-points, and ISO9886 14-points [26]. Based on Hardy and DuBois's 7-point formula [27], the weighting factors of different body regions were determined in the present study as listed in Table 1. The skin temperature of different regions in the body was measured using thermistors (N543R, Nikkiso-Therm Co., Tokyo, Japan).

**Table 1.** Weighting factors for the mean skin temperature of different body regions.

| Body Region $i$ | Weighting Factor $F_i$ |
|---|---|
| 1: Forehead | 0.07 |
| 2: Abdomen | 0.35 |
| 3: Forearm | 0.14 |
| 4: Back of hand | 0.05 |
| 5: Thigh | 0.19 |
| 6: Leg | 0.13 |
| 7: Back of Foot | 0.07 |

In this study, the human thermal load method was improved for assessing nonuniform environments in the up and down direction, such as on pavements. In general, global solar radiation influences the outdoor thermal environment. To evaluate the effects of climatic

radiation and radiation from the ground surface separately, net radiation is divided into downward net radiation $R_{net}\downarrow$ and upward net radiation $R_{net}\uparrow$ as

$$R_{net}\downarrow= (1-\alpha_h)R_{sh}\downarrow+\varepsilon_h\left\{R_{ln}\downarrow-\sigma(T_{skin}+273.15)^4\right\} \tag{8}$$

$$R_{net}\uparrow= (1-\alpha_h)R_{sh}\uparrow+\varepsilon_h\left\{R_{ln}\uparrow-\sigma(T_{skin}+273.15)^4\right\} \tag{9}$$

where $\downarrow$ indicates the downward components and $\uparrow$ indicates the upward components. To represent the effect of the direction of irradiation and the size of the person on the net radiation, the person was simplified to a floating rectangular shape facing the sun, as shown in Figure 2. Upward and downward solar radiation and infrared radiation were then calculated as

$$R_{sh}\downarrow= \frac{A_{up}S\downarrow+A_f(S_T cosz+\gamma S_D)+\gamma S_D(A_l+A_r+A_{bk})}{A_D} \tag{10}$$

$$R_{sh}\uparrow= \frac{A_{bt}S\uparrow+\gamma S\uparrow\left(A_f+A_l+A_r+A_{bk}\right)}{A_D} \tag{11}$$

$$R_{ln}\downarrow= \frac{A_{tp}L\downarrow+\gamma L\downarrow\left(A_f+A_l+A_r+A_{bk}\right)}{A_D} \tag{12}$$

$$R_{ln}\uparrow= \frac{A_{bt}L\uparrow+\gamma L\uparrow\left(A_f+A_l+A_r+A_{bk}\right)}{A_D} \tag{13}$$

where $A_{up}$, $A_f$, $A_r$, $A_l$, $A_{bk}$, and $A_{bt}$ (m$^2$) denote the areas of the upper, front, right, left, back, and bottom planes, respectively. $L\downarrow$ and $L\uparrow$ (W/m$^2$) represent the infrared radiation from the sky and from the ground measured by the net radiometer. $S_T$ (W/m$^2$) denotes the direct solar radiation and $S_D$ (W/m$^2$) denotes the diffuse solar radiation; they were estimated using Udagawa's formula [28] from $S\downarrow$ measured by the net radiometer. $z$ denotes the solar altitude angle (°), and $\gamma$ represents the view factor between the human and the sky or the surface, which was assumed to be 0.5 for an open area. The area ratio for each body plane was uniformly determined for each participant: $0.3A_D$ for $A_f$ and $A_{bk}$, $0.15A_D$ for $A_l$ and $A_r$, $0.05A_D$ for $A_{up}$ and $A_{bt}$, respectively.

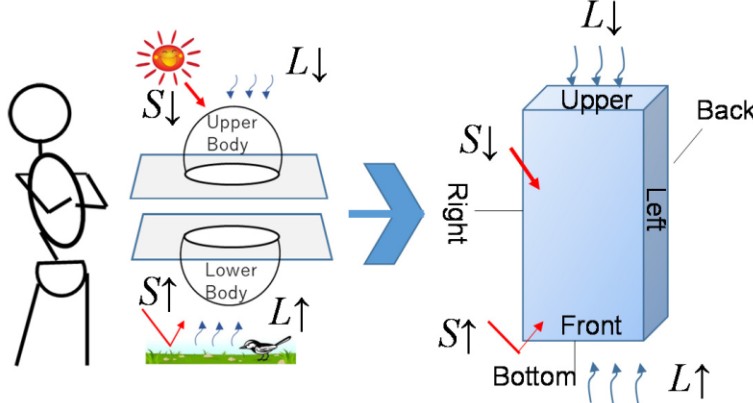

**Figure 2.** Abstraction of the separation of the upward and downward radiation components.

Convective heat loss $C$ is the sum of the dry heat loss from the skin ($C_{sk}$), and through the respiratory system ($C_{res}$). $C_{sk}$ is based on Burton and Edholm [29], and $C_{res}$ is based on the ASHRAE model [23]; the total convective heat loss is determined as:

$$C = C_{sk} + C_{res} \tag{14}$$

$$C_{sk} = F_{cl}h_c(T_{skin} - T_{air}) \tag{15}$$

$$C_{res} = 0.0014M(34 - T_{air}), \tag{16}$$

where $T_{air}$ (°C) denotes the temperature of the ambient air, $F_{cl}$ represents the Burton's reduction factor, and $h_c$ (W/m$^2$/°C) indicates the convective heat transfer coefficient [30] given as:

$$F_{cl} = \frac{1}{(1 + 0.209h_cI)} \tag{17}$$

$$h_c = 3.86 + 6.96v^{1.02} \tag{18}$$

where $I$ (clo) denotes the clothing insulation [31] and $v$ (m/s) denotes the wind speed.

Evaporative heat loss $E$ is the sum of the wet heat loss from the skin ($E_{sk}$) and the respiratory system ($E_{res}$). The evaporative heat loss from the skin is a combination of thermo-regulatory sweating $E_{rsw}$ and insensible natural diffusion $E_{dif}$ [23].

$$E = E_{sk} + E_{res} \tag{19}$$

$$E_{sk} = E_{rsw} + E_{dif} \tag{20}$$

The amount and timing of evaporative heat loss is important; however, in practice, it is not easy to measure both in outdoor field experiments. Based on the two-node model [32,33], the evaporative heat loss is determined as

$$E_{rsw} = c_{sw}(T_b - T_{b,set})exp\left(\frac{T_{sk} - T_{sk,set}}{10.7}\right) \tag{21}$$

$$E_{dif} = 0.06\, E_{max} \tag{22}$$

$$E_{max} = LwF_{cl}h_c(P^*_{skin} - P_{air}) \tag{23}$$

$$E_{res} = 0.0173M(5.87 - P_{air}) \tag{24}$$

where $c_{sw}$ (W/(m$^2$K)) is a proportional constant for sweat control (=170), $T_b$ (°C) denotes the blood temperature, and $T_{b,set}$ and $T_{sk}$, (°C) denote the sweating threshold for blood and skin temperature, respectively. $Lw$ (°C/kPa) represents the Lewis ratio, $P^*_{skin}$ (kPa) denotes the saturated water vapor pressure at skin temperature, and $P_{air}$ (kPa) represents the water vapor pressure in ambient air. Regardless of the situation for thermo-regulatory sweating, the ratio of the diffusion evaporative heat loss was set to 6% of $E_{max}$. $E_{rsw}$ can be predicted as a function of blood temperature and mean skin temperature. The blood temperature was determined as:

$$T_b = (1 - \alpha_{sk})T_{cr} + \alpha_{sk}T_{skin} \tag{25}$$

where $\alpha_{sk}$ denotes the mass ratio of the skin component to the whole body. $T_{cr}$ (°C) denotes the core temperature of the human body, and the rectal temperature was measured using a thermistor (N543R, Nikkiso-Therm Co., Tokyo, Japan).

To understand the physiological conditions of the participants, the body-weight and heartrate were measured using a precise electronic balance (GP-100K, A&D Co. Ltd., Tokyo, Japan) and a heartrate sensor (RS800CX Training Management System, Polar, Kempele, Finland). To limit the above evaporative heat loss $E$ compared to the participant experiments, we set an upper limit $E'$ to total evaporative heat loss based on the time-averaged sweat evaporation from the body-weight measurement as:

$$E' = E_{sk}' + E_{res}' = \frac{l}{A_D}\frac{\Delta w}{t} \tag{26}$$

where $\Delta w$ (kg) denotes the body-weight change before and after the experiment, and $t$ (s) represents the duration between the body-weight measurement before and after the experiment. When $E > E'$, $E$ was replaced by $E'$.

2.2.5. Human Perceptions

As the perception of the participants' thermal comfort is a basic concept of environmental evaluation using the adaptive approach, we asked the participants to evaluate their perceptions on the thermal sensation, wettedness, and thermal comfort using the visual analog scale (VAS). Then, for quantification, each perception was scored from $-3$ to $3$ for thermal sensation, from $-2$ to $+2$ for wettedness, and from $-2$ to $2$ for thermal comfort, with linear interpolation based on the standards from the Architectural Institute of Japan [34]. The obtained scores correspond to the numbers listed in Table 2. Every perception was recorded on-site at 5 min intervals.

**Table 2.** Quantification of human thermal environment evaluated by the study participants.

| Value | Thermal Sensation | Wettedness | Thermal Comfort |
|-------|-------------------|------------|-----------------|
| 3 | Hot | - | - |
| 2 | Warm | Wet | Comfortable |
| 1 | Slightly warm | Slightly wet | Slightly comfortable |
| 0 | Neutral | Neutral | Neutral |
| $-1$ | Slightly cool | Slightly dry | Slightly uncomfortable |
| $-2$ | Cool | Dry | Uncomfortable |
| $-3$ | Cold | - | - |

2.2.6. Data Analysis

As the experiments were conducted outdoors, the variations in the ambient atmosphere could have affected the results. The climate was relatively stable during the 30 min of the experiment, and human thermal states were almost stable. Thus, the investigation was performed using the mean value recorded during the experiments with the participants. A Student's *t*-test analysis was conducted to investigate whether the difference in pavements in the variables was significant. The effect size for a *t*-test was evaluated using Cohen's d. The items measured and accuracy of the instruments are presented in Table 3 for uncertainty evaluations. Since the uncertainty of solar and infrared radiation measurement under outdoor conditions depends on many factors, please refer to the specifications for a more complete description (https://eko-usa.com).

**Table 3.** Measurement items and instruments.

| Parameter | Accuracy | Instrument |
|-----------|----------|------------|
| Air temperature | $\pm0.3$ °C (0–50 °C) | TR-73U, T&D |
| Relative humidity | $\pm5\%$ R.H. (10–95% R.H.) | TR-73U, T&D |
| Wind speed | $\pm2\%$ or 0.02 m/s | Model-6531, Kanomax |
| Surface temperature | $\pm1.5$ °C ($-40$–375 °C) | Thermocouple, J-type |
| Ventilatory gas | $\pm0.1\%$ for $O_2$/ $\pm0.2\%$ for $CO_2$ | VO2000, MGC Diagnostics |
| Body temperature | $\pm0.2$ °C (0–70 °C) | N543R, Nikkiso-Therm |
| Body weight | $\pm10$ g | GP-100K, A&D |
| Heartrate | $\pm1\%$ or 1 bpm | RS800CX, Polar |

**3. Results**

*3.1. Experimental Conditions*

The air temperature, humidity, solar and infrared radiation, and wind speed measured at a 1.5 m height during the experiment are shown in Figure 3. The average values (mean $\pm$ s.d.) of these physical quantities measured over AS were as follows: air temperature ($33.3 \pm 1.4$ °C), humidity ($53.2 \pm 7.0\%$ R.H.), global solar radiation ($583 \pm 237$ W/m$^2$), infrared radiation from the sky ($520 \pm 13$ W/m$^2$), and wind speed ($0.79 \pm 0.18$ m/s). The average maximum temperature in late August in Okayama is 32.7 °C, based on the database from the Japan Meteorological Agency. The global solar radiation showed some fluctuation

because of the condition of cloud cover. However, other variables were mostly stable. Thus, the overall climatic conditions during the experiment can be considered as typical of summer weather.

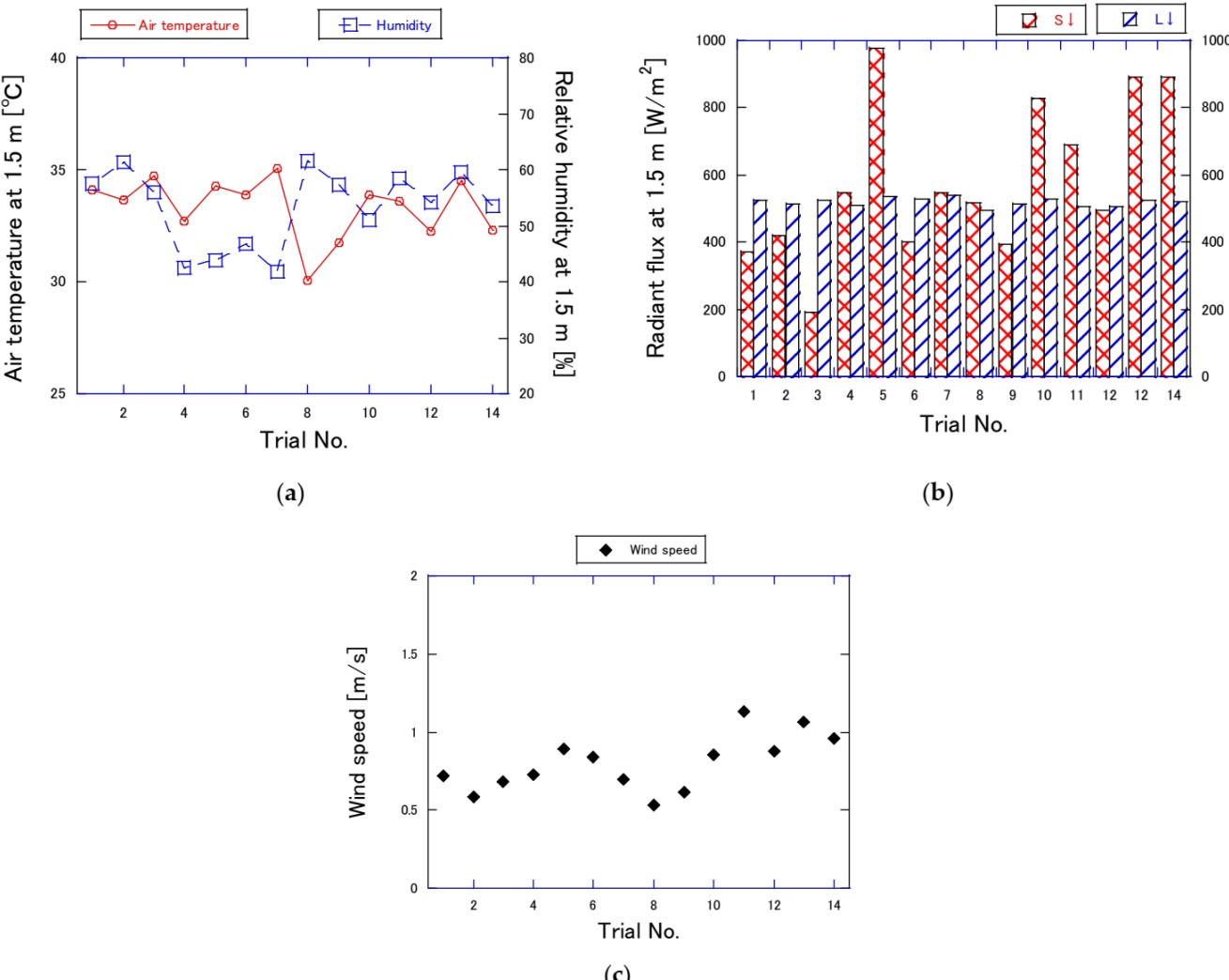

**Figure 3.** Climatic conditions at 1.5 m during each trial of the experiment: (**a**) air temperature and humidity, (**b**) global solar radiation and infrared radiation from the sky, and (**c**) wind speed.

### 3.2. Pavement Conditions

The surface temperatures of both pavements were compared using infrared images, as shown in Figure 4. The surface temperature was uniformly distributed over the surface. Using the thermocouples, the average surface temperature of AS and WR during the experiments was measured to be $50.2 \pm 5.0$ °C and $40.7 \pm 2.9$ °C, respectively, and this difference was found to be significant ($p < 0.001$, d = 2.38). The performance of WR is analyzed in Figure 5. The surface temperature difference varied approximately linearly with the surface temperature on AS, so the effectiveness of surface temperature reduction could be assessed completely.

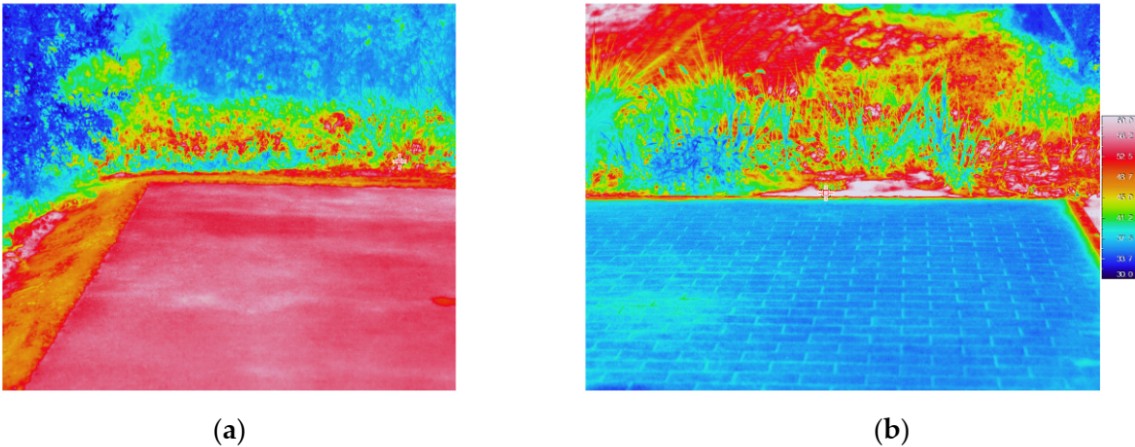

**Figure 4.** Thermal images of both pavements (at 10:20 a.m. on 26 August 2016): (**a**) AS and (**b**) WR.

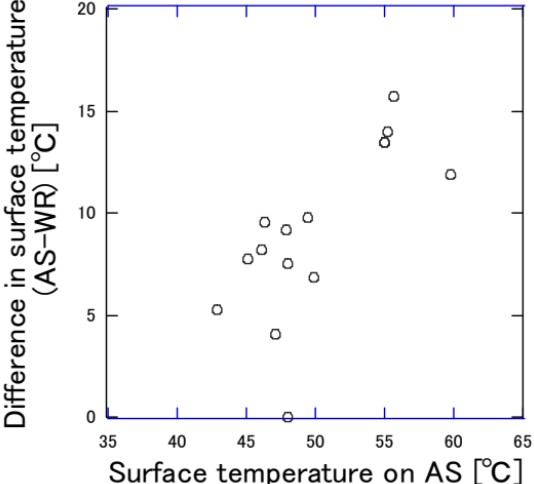

**Figure 5.** A comparison of surface temperatures.

The vertical temperature and humidity profiles of both pavements are shown in Figure 6. To eliminate the effect of temperature on humidity, the absolute humidity is presented in the figure. It is natural for the air temperature to be lower as the measuring point increases. The air temperature profiles showed similar values for both pavements, except on the surface. Absolute humidity fell within the range of approximately 17 g/m$^3$. A clear effect of water evaporation on WR and height on absolute humidity could not be observed.

The values of albedo were obtained with acceptable fluctuation, as listed in Table 4. The measurement was conducted simultaneously during participant experiments to assess the WR effect at the wet state. AS absorbs more solar energy and WR reflects more solar energy because pavements are considered as opaque and AS has a lower albedo value.

**Table 4.** Reflectance of pavements.

|  | AS | WR |
|---|---|---|
| Reflectance [1] | $0.085 \pm 0.010$ | $0.253 \pm 0.029$ |

[1] Values in mean ± s.d.

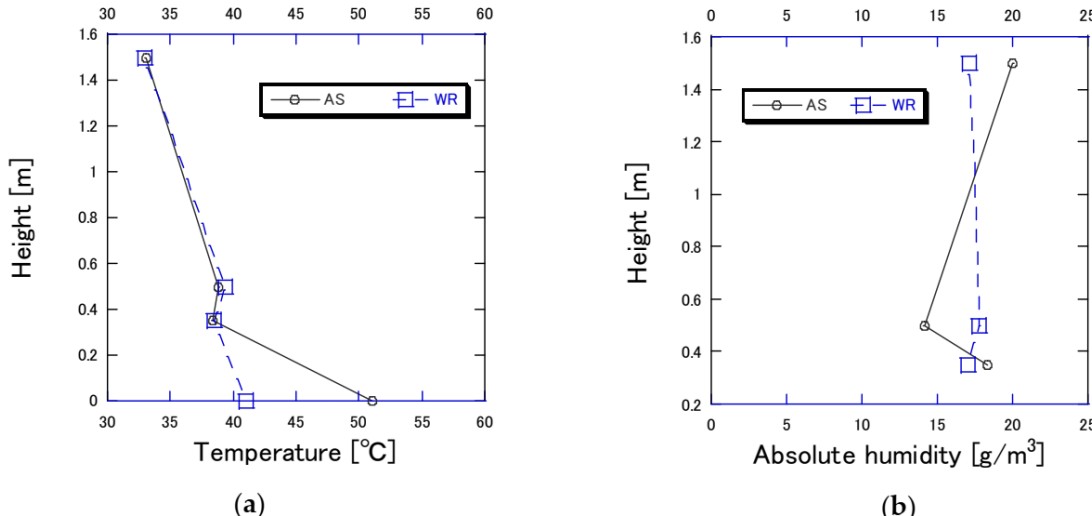

**Figure 6.** The vertical profile over the pavements: (**a**) temperature and (**b**) absolute humidity.

### 3.3. Human Thermal Condition at a Height of 1.5 m

The human thermal loads at pedestrian height (1.5 m) for each trial, the relative contribution of $R_{net}\uparrow$ of AS and WR from $S\uparrow$ and $L\uparrow$, and the mean net radiations are compared in Figure 7, and each thermal load component is summarized in Table 5. Since $F_{load} = 0$ is thermally neutral, it is assumed to be comfort. In summer, a lower human thermal load is preferred. Since each participant experienced environments on both pavements, the metabolic rate should be similar (87 W/m$^2$). Air conditions such as temperature, humidity, and airflow were similar for both pavements and did not affect heat loss. Thus, other factors such as personal differences could have affected the difference in heat losses. A noticeable difference was observed in the net radiation. $R_{net}\downarrow$ is dependent on climatic radiation and was similar for both pavements, while $R_{net}\uparrow$ indicates a smaller tendency for WR ($p = 0.10$, d = 0.31). Further, the reflected solar radiation $S\uparrow$ of WR was significantly larger ($p < 0.001$, d = 2.31) and the infrared radiation $L\uparrow$ of WR was significantly smaller than that of AS ($p < 0.001$, d = 2.39). The total human thermal load was measured to be 237 ± 38 W/m$^2$ for AS and 215 ± 49 W/m$^2$ for WR. A significant difference was observed in the human thermal load between AS and WR, and it presented a small sized effect ($p = 0.01$, d = 0.47).

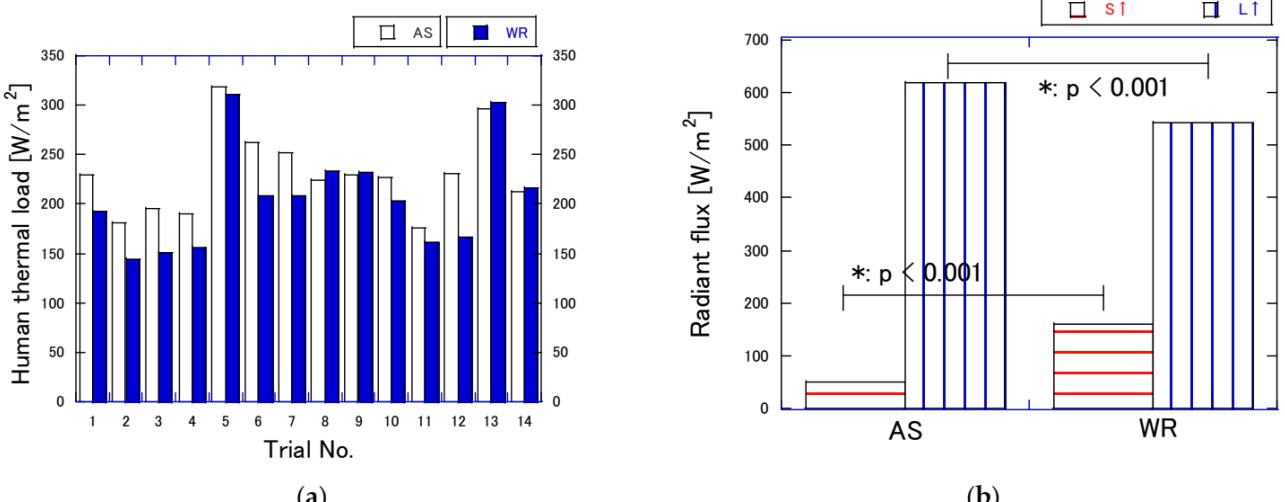

**Figure 7.** *Cont.*

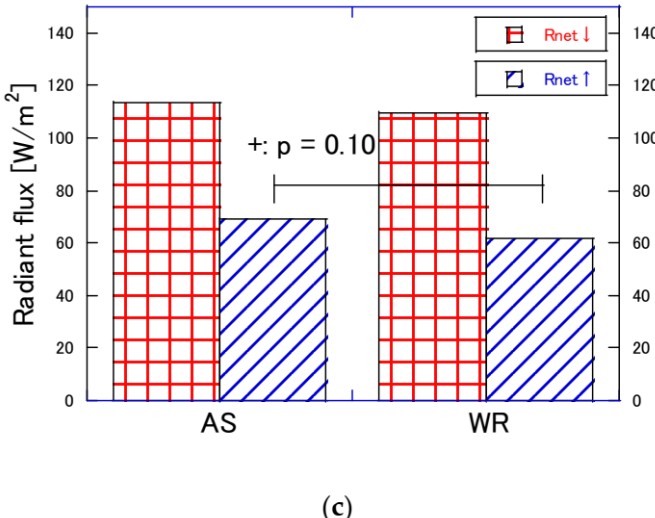

(**c**)

**Figure 7.** Comparison between AS and WR at pedestrian height (1.5 m): (**a**) human thermal load, (**b**) comparison of radiation components of AS and WR, and (**c**) net radiation.

**Table 5.** Components of human thermal load.

| Component | AS | WR |
|---|---|---|
| Metabolic rate | 87 ± 2 | 87 ± 2 |
| Workload | 0 | 0 |
| Net radiation | 193 ± 42 | 181 ± 55 |
| Convective heal loss | 15 ± 6 | 18 ± 4 |
| Evaporative heat loss | 29 ± 3 | 35 ± 5 |
| Human thermal load * | 237 ± 38 | 215 ± 49 |

* Significant difference ($p$ = 0.01, d = 0.47).

### 3.4. Human Perceptions

The mean value of each perception is shown in Figure 8. In the case of AS, the thermal sensation was +0.69, wet sensation was +0.04, and comfort sensation was −0.33. In the case of WR, the thermal sensation was +0.48, wet sensation was −0.02, and comfort sensation was −0.14. The participants found that the environment was hot and uncomfortable overall. There was a trend towards reporting a cooler sensation on WR ($p$ = 0.04, d = 0.31), and WR did not significantly affect other perceptions. Interestingly, no difference in wet sensation was induced on WR, even when the participant stood on the wetted WR surface.

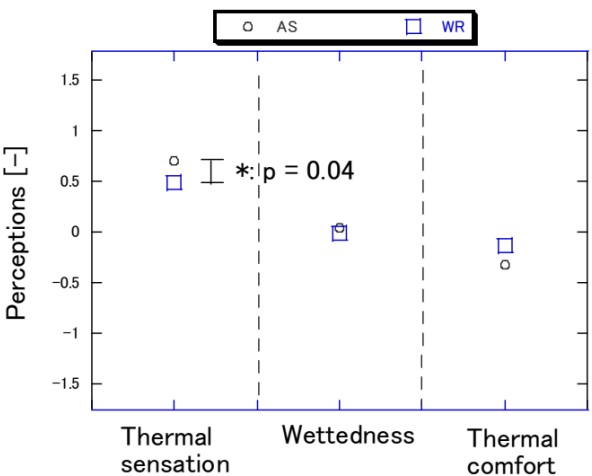

**Figure 8.** A comparison of the perceptions of the participants on AS and WR.

The relationships among the human thermal load, thermal sensation, and thermal comfort are plotted in Figure 9. Figure 9a shows the mean value for each variable (vertical and horizontal axis) and Figure 9b shows the line of equality ($y = x$). The human thermal load correlated with thermal sensation, and an almost linear relationship between human thermal load and thermal sensation can be observed. Overall, the plots on WR tended to be distributed in regions of smaller human thermal load and cooler perception. Further, thermal comfort had a correlation with thermal sensation (Figure 9c), as is often reported by researchers.

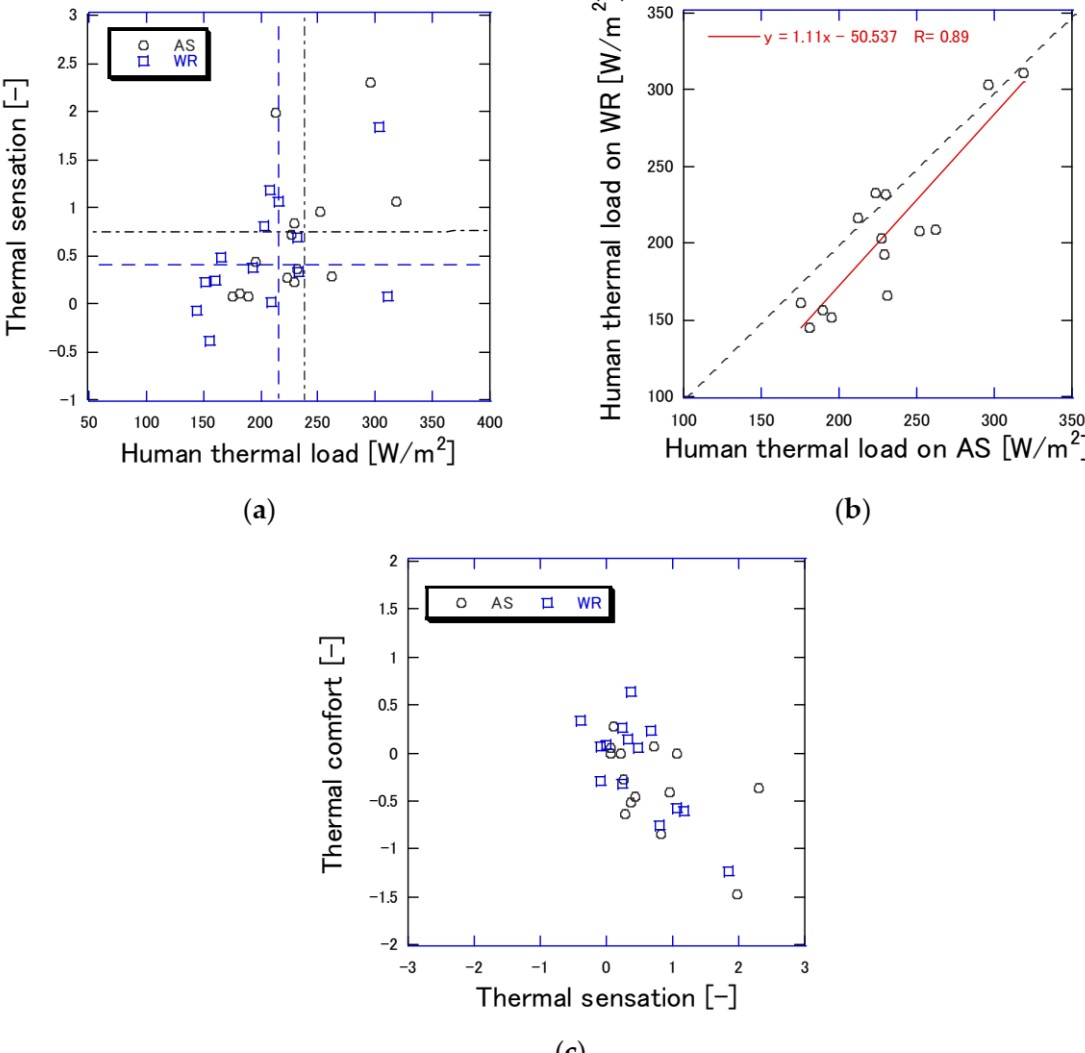

**Figure 9.** Relationships among the human thermal load, thermal sensation, and thermal comfort: (**a**) human thermal load and thermal sensation on AS and WR, (**b**) comparison of human thermal load of AS and WR, (**c**) thermal sensation and thermal comfort on AS and WR.

## 4. Discussion

### 4.1. Effects of Properties of WR on Human Thermal Environment

Two types of cool pavements are considered for surface improvement: reflective and evaporative. The WR in this experiment had a water-retentive function and higher reflectivity (=0.253) relative to AS (=0.085); thus, it is a hybrid cool pavement. The surface temperature on WR was significantly lower (by 9.5 °C) than conventional AS, and the potential of surface temperature reduction tended to be larger as the solar radiation became intense. However, the air temperature and humidity above both pavements were not

significantly different. Numerous studies have reported on the effect of the color of the pavement materials on their surface temperature, and these effects can lead to a reduction of up to approximately 20 °C, depending on material properties, climatic conditions, and timing [35]. The air temperature was reported to be less sensitive than surface temperature [7]. There was a concern that WR would induce wet sensation; however, the evaporation above WR did not make the pedestrians on it feel wet. The areas of the test pavements used in this study may not be sufficient to influence the ambient environment, and a vast pavement may be more influential on temperature and humidity formation. If the lower surface temperature is widely spread on the pavement, it can potentially suppress the air temperature, which would be useful in mitigating the effects of UHIs. Such pavement modifications will be studied in future work.

### 4.2. Human Thermal States and Perceptions

The environmental measurement of pavements has been widely conducted. However, subjective evaluation of functional pavements such as WR by humans has rarely been reported previously. Using the indicator of human thermal load and its components, we compared and evaluated the thermal conditions and human effects of the WR pavement.

The sensible and evaporative heat loss did not show a significant difference because air temperature, humidity, and wind conditions experienced by the participants on both pavements were similar. The net radiation consists of the effects of solar radiation and infrared radiation. Because $R_{net}\downarrow$ is dependent on climatic radiation, $R_{net}\uparrow$ can reflect pavement effects. In Figure 7b, the reflected solar radiation $S\uparrow$ of WR was significantly larger and the infrared radiation $L\uparrow$ of WR was significantly smaller than that of AS. As the infrared radiant emission is proportional to the fourth power of the temperature, the infrared radiation from WR was lower as the surface temperature level was lower. It is likely to have had a negative impact on the participants due to the increased reflected solar radiation, especially under strong solar radiation. This suggests that an optimum reflectance value for reducing $R_{net}\uparrow$, namely human thermal load, should be examined for environmental design.

To explore the degree of adaptation to the environment based on the proposed approach, the relationships among the human thermal load, thermal sensation, and thermal comfort are examined in Figure 9. The human thermal load correlated with the thermal sensation, and the thermal sensation correlated with the thermal comfort. The plots on WR tended to be distributed in regions of smaller human thermal load and cooler perception. In addition, a linear relationship between the human thermal load on AS and WR is observed in Figure 9b. The slope of the regression line in Figure 9b is slightly larger than 1.0. This indicates the possibility that the difference in the human thermal load between AS and WR could become narrower for larger human thermal load conditions, because under these conditions, solar radiation should be intense, and thus, humans would receive more reflected solar radiation on the WR pavement. Since thermally neutral states can be considered as thermal comfort, the relationship between thermal sensation and thermal comfort shows a mound-shaped distribution. Thermal comfort has a negative correlation with thermal sensation in Figure 9c due to the warmer climate that was present during the experiment.

### 4.3. Limitations

This study was designed to create thermal comfort for outdoor pedestrians based on a field experiment on WR. The outdoor space varies both spatially and temporally, and thus it may be important to develop a thermal comfort evaluation method that considers the instantaneous and local effects of thermal comfort or discomfort. A line of the best regression fit for scatter plots is useful for drawing predictions. However, all experiments were conducted in a warm environment in the present study. Hence, in future work, we propose to perform additional studies in a variety of environments, such as in more neutral and cooler environments. Since the data varied because of exposure to environmental conditions caused by the outdoor participant experiments, participants potentially reacted

physiologically differently, and reported a somewhat wider range of thermal perceptions. Further, it is necessary to understand the effect of human activities such as street walking on the metabolic rate and clothing functions instead of when the pedestrian is stationary, for future work. Since evaporative heat loss is an important factor considering thermal comfort under heat, it is desirable to utilize a novel on-site measuring system for sweat evaporation. Further studies are required to optimize and modify pavement performances such as reflectance by considering the relative contribution of radiation components. The properties of pavements, such as emissivity and reflection, also need to be investigated in association with conditions of pavements such as surface roughness. Although the measurement was conducted at a height of 1.5 m, the effect of the height above the pavement on the thermal environment was important to be examined. Human thermal load is a concept for quantifying the human thermal environment in the form heat flux, and the determination method for each term can be replaced, modified, or updated according to the requirement. To solidify the findings of this study, comparisons with outdoor thermal indices such as PET and UTCI can be conducted as a future work.

## 5. Conclusions

Currently, there are few established methods that allow easy assessment of outdoor thermal comfort specific to the evaporative-pavement-occupied pedestrian environment. In this paper, we proposed quantification techniques to study the influence of WR pavements and conventional AS pavements on the thermal environment, human body, and thermal comfort; in particular, we focused on the relative contribution of radiation components.

The key experimental observations on the thermal behavior of WR are as follows: (1) WR significantly reduced the surface temperature when compared to AS. However, the air temperature and humidity above both the pavements were not significantly different. (2) The sensible and evaporative heat loss of humans did not make a significant difference due to the similar climatic air conditions. The reflected solar radiation from WR was significantly larger and the infrared radiation of WR was significantly smaller than that of AS. Thus, WR significantly reduces human thermal load. (3) The proposed human thermal load correlates with the subjective thermal sensation. WR tends to induce a cooler sensation and does not induce an unpleasant wet sensation. As there is a strong relationship between thermal sensation and thermal comfort, we can utilize human thermal load as an environmental indicator and assessment tool for UHI adaptation. (4) It can be concluded that the thermal environment of WR is better in terms of human thermal state when compared to AS. However, there is still room for improvement of the reflectance of WR by reducing the reflective solar heat received. Experiments in more realistic scenarios, such as when walking on the street, should be performed.

**Author Contributions:** Conceptualization, Y.S., M.A.; methodology, validation, and investigation; Y.S., M.A., J.N.; data curation, H.O.; writing—original draft preparation, Y.S.; supervision, A.Y. All authors have read and agreed to the published version of the manuscript.

**Funding:** This work was partly supported by JSPS KAKENHI Grant number 26281057 (P.I. Atsumasa Yoshida).

**Institutional Review Board Statement:** The study was conducted according to the guidelines of the Declaration of Helsinki, and retrospective approved by the Ethics Committee of Osaka Prefecture University (Project title: The relationship between thermal environmental factors and human physiological responses (PI: Atsumasa YOSHIDA), approved date: 30 September 2016).

**Informed Consent Statement:** Informed consent was obtained from all subjects in-volved in the study.

**Data Availability Statement:** Data sharing not applicable.

**Acknowledgments:** The authors acknowledge Yayoi Satsumoto of Yokohama National University and Tomonori Sakoi of Shinshu University for their technical advice.

**Conflicts of Interest:** The authors declare no conflict of interest.

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
