# Peer review of "Experimental Determination of Pedestrian Thermal Comfort on Water-Retaining Pavement for UHI Adaptation Strategy"

_atmosphere, doi:10.3390/atmos12020127_

Round 1

Reviewer 1 Report

The structure of the paper and the results were already good and now the the manuscript has been really improved by means of more clear explanations and extensive rewriting in all the sections. The results The revised paper deserves to be published.

Author Response

We thank you for your comments, which have helped us to improve the manuscript.

Reviewer 2 Report

I do not feel the authors have made any substantial changes to the work. The fundamental issues still remain.

The authors seem to be missing a fundamental aspect regarding human thermal load – it is not a common English term and should be defined before first use – and an abstract must be able to stand on its own, without relying on the full article for explanations. Authors explain it up ahead in Section 2.2.4

Section 2.2.2 – again the 1.5 m height is odd. ASHRAE recommends certain heights based on centre of gravity of a person or breathing height or ankle height. 1.5 does not correspond to any of this.

Why is equation 4 provided since authors do not actually measure VO2 or VCO2?

Figure 9 c is trivial – it is an often reported relation, without much practical value. I do not see the point of including it here

Since you already have the measured environmental parameters, why not include PET and UTCI values in this work? It does not make sense to have it as a future work – it is not a substantial addition

Similarly, a simulation with the measured conditions and a higher albedo surface could provide an idea on how these novel surfaces perform vis a vis surfaces with higher albedo.

Author Response

Thank you again for providing your valuable suggestions and the insightful comments on improving our research. The revisions are highlighted in red color in the revised manuscript.

  1. Human thermal load

As we mentioned in the manuscript, “human thermal load” is a measure for evaluating human thermal environment outdoors based on the energy balance of human. We added the explanation in abstract section, ahead in Section 2.2.4.

  1. Height of the measurement

We agree with the reviewer that the height of measurement is important for the outdoor environmental assessment. This was already mentioned in the manuscript. Because previously studied were conducted at 1.5 m height for pedestrian comfort, we also conducted 1.5 m height for the measurement. The experiment involving the effects of heights on the human thermal environment is planned as a future work. This was already stated in the limitations in the manuscript.

  1. Equation (4)

As readers can see in Figure 1, we do actually measure VO2 or VCO2. However, in some occasions, the resting metabolic rate is preliminary determined in an indoor chamber using Eq. (4), and then, the metabolic rate is determined by multiplying with the met value for convenience.

  1. Figure 9 c

Yes, it is an often reported relation. We could obtain the same relation, and we believe that this may validate our results.

  1. Comparison with other indices such as PET and UTCI

We agree the reviewer that comparison with other outdoor thermal models such as PET, UTCI are very beneficial of this issue. The experiment involving the effects of height, human activity etc on the human thermal environment and the comparison with other indices is reported as a future work.

  1. Albedo modification

As we stated in the manuscript, further studies are required to optimize and modify pavement performances such as reflectance. This could be achieved by considering the relative contribution of S and L components.

Reviewer 3 Report

The paper is well thought off and organised. The subject matter is important and my overall assessment of the manuscript is positive. I would recommend to publish the paper because it contains interesing data and findings, however I believe that the manuscript needs to be improved. I have provided the following comments:
- The introducion provides the general background of the study and the importance of the subject matter, which is fine. However, tha Authors should clearly state the novelty of their paper - the manuscript is focused on the thermal environment on a water retaining and conventional pavements. Please provide more details on this very issue and data available in literature. It is slightly mentioned with only 2 sentences at the end of the paper (lines 396-399) - but I believe that it should be extended and placed in the introductiory part.
- The paper is an experimental study and should contain the error analysis of the thermal load calculations (the thermal loads are even presented in the abstract due to their importance as 237 W/m2 for AS and 215 W/m2 for WR - but it actually is 237 +/-...error... and 215 +/- ...error...)
- line 68-70: when abreviations are used for the first time (PET, UTCI, PMV), they should be given in full, for example: "PMV (Predicted Mean Vote)"
- line 114: apart from providing height and weight separately, the range of BMI (body mass index) might be added here.
- chapter 2.2.2 - A table with measured values (air temperature, humidity and others) might be welcome here, containing measurement ranges and errors (for example +/- 0.4oC for temperature, and etc) - the same applies to measurements of surface temperatures (line 137) with thermocouples, where measurement errors are not given (as well as in the case of thermistors in line 203).
- line 191: can the Authors provide reference (a textbook/paper) for reflectance and emissivity (not "emittance") values?
- line 316 (Fig. 4). The surface characteristics of WR and AS are undoubtedly different - thus, the emissivity values (which are input into the infrared camera software) mush have been different. What were the emissivity values used? Where they measured onsite or taken from literature?
- line 323: surface roughness might also be a factor worth consideration. Have there been any measurements of surface roughness of these two pavement types?
- line 350: human thermal load for AS was usually larger than for WR (which is understandable), but in some case it is not (no 8,9,13,14). Can the Authors provide any explanation? The Authors have not provides any error analysis regarding the calculation of thermal load - maybe the difference fall into the error band and the results for AS are actually larger than for WR (those differences are small enough to back such a statement, but without the error analysis it is just an assumption)
- line 370 (fig. 9a). I would suggest to add two trendlines and the R2 value (for AS and WR) to investigate how strong the relation betwwen thermal load and thermal sensation it.
- ref. 27: standard ISO 9920:2007 has been replaced with ISO 9920:2009

Author Response

Thank you for providing your valuable suggestions and the insightful comments on improving our research. The revisions are highlighted in red color in the revised manuscript.

  1. Introduction (novelty of their paper)

We agree with the reviewer that the novelty of the study should be clearly presented. There are mainly reflective and evaporative pavements for UHI mitigations. Enhancing the surface reflectivity would be the mainstream, however, we believe that water retaining pavements are human and environmentally friendly. Basically few studies have focused on the effects of human aspects on functional evaporative pavements and how humans respond to these pavements. One aim of the present study is to assess the human thermal environment on a functional pavement using the quantitative human thermal load calculated from the environment and the human variables, especially focusing on pedestrian space. We also investigated individual thermal perceptions simultaneously.

  1. Error analysis for the human thermal load

We agree with the reviewer that error analysis should be conducted properly. Values in mean±s.d. was now presented in the manuscript and thus the total human thermal load was measured to be 237±38 W/m2 for AS and 215±49 W/m2 for WR.

  1. Abbreviations

Abbreviations were explained when they were used for the first time to enhance clarity.

  1. BMI

We calculated and added the Body Mass Index (BMI) to categorize our participants. The mean BMI value was 21.1±1.7 and the value were categorized into the healthy weight range.

  1. Measurement instruments

It is important to understand the instrument specifications. Thus, the items measured and the instruments with its accuracy are now listed in a table. We selected instruments for the accuracy and the applicability of outdoor experiments.

  1. Reflectance and emissivity

The radiative properties of skin generally varies depending on skin tones. The values of reflectance (about 0.3) and emissivity of human body (0.95-1.0) were determined based on the literatures.

  1. Surface characteristics

It is important to set the proper emissivity of surfaces for the infrared camera, however, the actual emissivity values are not decided. Since the emissivity is usually more than about 0.95 for pavements, the emissivity was set to 1.0 for both pavements. The reflection of pavements may be different depending on composition, structure, age, and so on. Our materials have relatively rough surfaces and measurement was conducted using larger spaces, the effects of the reflection properties of a pavement surface on the results would be limited. Totally, the properties of pavements such as the emissivity and the reflection need to be investigated in association with conditions of pavements such as surface roughness.

  1. Analysis using the human thermal load

We agree with the reviewer that error analysis should be conducted properly. Values in mean±s.d. was now presented in the manuscript.

The human thermal load can reflect not only physical environments but also physiological states of humans. Since the data varies because of exposure to environmental conditions caused by the outdoor participant experiments, participants potentially react physiologically differently. As we stated in the manuscript, personal differences may affect the difference in heat losses and this is one of the reason why human thermal load for WR was occasionally larger than for AS.

  1. Fig. 9 a

It would be a good idea to add two trend lines and the R2 values for Fig. 9 a. Considering limitations, we are planning to investigate the relation between thermal load and thermal sensation after additional experiments in various situations.

  1. ISO9920

Thank you for pointing this out. We indeed utilized ISO 9920:2007. ISO 9920:2009 will be referred in the future research.

Reviewer 4 Report

The manuscript is really interesting and deals with a very important issue of the urban context for the adaptation of the city in assuring a better wellness to population.

The introduction is complete and clearly presented. It results immediatly clear the objective of th study and the possible applications.

The experiment from a physical-physiological point of view is perfectly described. About the participant reactions set-up I must admit I'm not fully entitled to say if the set-up is properly arranged, it appears to me reasonable but I'm not an expert to understand if the trial is correct. As an example, I do not know if the term "psycological reaction" is well poses into this context (2.2.1)

Starting from point 2.2.2 up to 2.2.6, where I feel myself more confortable, the experiment properly reports all necessary passages in a clear and undestrandable way to a wide audience and represents a very uselful guide for scholars involved in the adaptation studies.

Assuming point 2.2.1 was properly conducted, the results are well presented and of very high interest. The figures reported are very indicative. The discussion is really well posed and the authors reports very clearly possible limitations to their study. The conclusion are properly drawn.

 Remarks:

lines 69-70: SET* and PMV must be accompained by their meaning.

Abstract: the abstract does not report that the study it is not only limited to the usual determination of wellness conditions but it also report an experimental part linked with "perceptions" in terms of individual response to the experiment. In my opinion it is necessary to clarify it including a short indication.

Given my limitation in reviewing paragraph 2.2.1, in my opinion the paper is of marked interest and I suggest it for publication after minor corrections.

Author Response

Thank you for providing your valuable suggestions and the insightful comments on improving our research. The revisions are highlighted in red color in the revised manuscript.

  1. SET* and PMV

Abbreviations were explained when they were used in the first time. The standard effective temperature (SET*) and the predicted mean vote (PMV) are broadly used thermal comfort indices. Outdoor indices such as PET and UTCI were also spelled out.

  1. Abstract (perceptions)

We asked individual thermal perceptions simultaneously during participant experiments. We added this in the manuscript.

Round 2

Reviewer 2 Report

I am unsure that this work should be published without a comparison of the index put forth by the author to more established indices like PET/UTCI - to actual values.

Author Response

We agree that the comparison with other indices would be interesting to know. In this paper we prefer to focus on the situations on pavement especially for WR, and so we will leave the comparison with other indices and additional measurements to other research. This explanation has been added in the limitation section. Thank you again for the suggestion.

Reviewer 3 Report

The paper has been improved, however the Authors have not made any changes to the Introduction regarding my first comment on the novelty and the state-of-the-art (extended literature study), despite the fact that the Authors adressed this comment in their message to the reviewer. I still think that this needs to be done in the final version of the paper (novelty is mentioned with only 2 sentences at the end of the manuscript: lines 396-399 - but I believe that it should be extended and placed in the introductiory part).

Author Response

We are sorry that our novelty and literature studies was not clearly mentioned in the manuscript.

Our study was designed to focus on the outdoor human thermal environment on a functional pavement relative to a conventional dense-asphalt pavement. A number of studies has addressed surface temperatures on “cool pavements” (We added a few references). Although the surface temperature reduction is important, this is one of factors that influences human thermal environment. In order to experimentally assess the human thermal environment, the human thermal load calculation was modified regarding radiative components and subjective thermal perceptions simultaneously analyzed in this paper.

Thank you again for the suggestion.

This manuscript is a resubmission of an earlier submission. The following is a list of the peer review reports and author responses from that submission.

Round 1

Reviewer 1 Report

The topic of the paper, regarding the mitigation of the heat island effect due to the asphalt pavement, is very interesting.

The scienfic approach is based on an original experimental approach, which has involved university students and full scale experimental pavements; analytical equations has been used fot the modeling of the phenomenon. 

The main results have outlined the water retaining method as effective in order to significantly  mitigate the heat island effect; such results are interesting and deserve to be published.

It is suggested to the authors to increase the size of figure 1 and to design a contour blackline in figure 4.

Reviewer 2 Report

Dear Authors, here below my observations about you research paper.

You will find other minor comments in the pdf marked file.

Best regards.

Introduction

  • There are no references to other similar studies and, the part of the manuscript devoted to the assessment of thermal comfort conditions (L60-66) could reflect a poor knowledge in the field of the ergonomics of the physical environment. Particularly, PMV and de Dear’s adaptive approach are specific for indoor environments, whereas no mention to other most effective metrics as UTCI or PET has been found.

Methods

  • Used method for assessing thermal load on the subject is based upon a model developed by you in 2011 that is quite unknown within the literature (only 25 citations from Scopus database). This reduces the significance of the used methodology.
  • The metabolic rate value used in this research (1,2 met) is well only for standing people. On the contrary, most people walk within a city with higher metabolic rate values.
  • Used formula for the skin temperature was developed in 1938 by Hardy and Du Bois. On the contrary more recent and standardized formulas exist (see ISO 9886). You are invited to explain the reason why of this uncommon choice
  • The evaluation of evaporative term is ambiguous. This is why some equations are referred to the two-node model whereas eq. (2) is for the body as a whole.
  • Clothing insulation values are not reported. In addition there is no mention about the effect of body movement and air action on clothing thermophysical properties (this sounds strange as you cite ISO 9920: 2007 Standard among references).

Results and discussion

  • You are invited to specify the effect of the height above the ground on simulations devoted to human thermal conditions. This is why air temperature, absolute humidity values vary vertically (see fig. 5) and it is reasonable that the height also affects IR radiation coming from the ground. This information is absolutely necessary because, from table 6, the thermal loads for AS and WR differ of about 10%. This is a very low amount if the uncertainties of the model are considered.

Discussion

  • Please change the name of this section if section 3 is named as “results and discussion”
  • You tried to discuss obtained results in term of adaptation phenomena. Well, in principle it is good, but from the analysis of your reasoning it seemed to me that you misuses the word “adaptation” (see De dear and Brager 1998 as in https://escholarship.org/content/qt4qq2p9c6/qt4qq2p9c6.pdf).

Bibliography

This section is not up to date and reveal a not adequate knowledge in the field of ergonomics of the physical environment. Thermal comfort objective assessment references are typical for indoor conditions and ASHRAE Handbook last version is 2017.

Reviewer 3 Report

The human thermal load was also found to be significantly different: 237 W/m2 for AS and 215 W/m2 for WR. – please better expound what you mean by thermal load here

Okayama is known as “land of sunshine” due to its minimal rain and mild climate throughout the year. – How do you source water for the pavements? This means a major departure from natural ground retaining rain water

I feel there should also have been an inclusion of high albedo pavements in the comparisons. Alternately, authors could do the physical measurements for the WR, without water addition. Just the physical and climatic measurements, not human participant studies.

What was the participants’ activity on the pavements? Were they just standing?

Section 2.2.2 – Authors should justify their choice of heights for the measurements – some suitable references maybe provided

I would suggest adding one or both of the widely used outdoor comfort indices UTCI and PET to the work. It would be great to have a comparison with the model that the authors have created for the net radiation load.

Fig 5 – are these measurement results with participants present or without them? Could presence of participants explain the kink in the humidity curve?

Table 3 – I am a little confused here and it maybe just me. Was the albedo value different because of water retention or because of the material difference? Was albedo for WR also measured before feeding it water?

Fig 6 b does not provide a convincing illustration that Rnet (up) was significantly different between the two cases. Authors need to provide p value (not just say p < 0.05) and effect sizes. Statistical significance need not always translate to and operational significance.

Section 3.4 – no wet sensation – please provide information on the footwear used

Please limit results to results section. Hence, the figures in Discussions should be moved to Results